

# Ensemble modeling of black pomfret (*Parastromateus niger*) habitat in the Taiwan Strait based on oceanographic variables

Sandipan Mondal[1,2], Ming An Lee[1,2,3], Yu-Kai Chen[4] and Yi-Chen Wang[1,2]

[1] Environmental Biology & Fishery Science, National Taiwan Ocean University, Keelung, Taiwan
[2] Center of Excellence for Ocean Engineering, National Taiwan Ocean University, Keelung, Taiwan
[3] Doctoral Degree Program in Ocean Resource and Environmental Changes, National Taiwan Ocean University, Keelung, Taiwan
[4] Coastal and Offshore Resource Research Center, Fisheries Research Institute, Council of Agriculture, Kaohsiung, Taiwan

## ABSTRACT

The location, effort, number of captures, and time of fishing were all used in this study to assess the geographic distribution of *Parastromateus niger* in the Taiwan Strait. Other species distribution models performed worse than generalized linear models (GLMs) based on six oceanographic parameters. The sea surface temperature (SST) was between 26.5 °C and 29.5 °C, the sea surface chlorophyll (SSC) level was between 0.3–0.44 mg/m$^3$, the sea surface salinity (SSS) was between 33.4 °C and 34.4 °C, the mixed layer depth was between 10 °C and 14 °C, the sea surface height was between 0.57 °C and 0.77 °C, and the eddy kinetic energy (EKE) was between 0.603 °C. According to the statistical findings, SST is merely a small effect compared to SSS, SSC level, and EKE in terms of impacting species distribution. By combining four effective single-algorithm models with no obvious bias, an ensemble habitat model was created. The ranges of 117°E–119°E and 22°N–24°N have the highest annual distributions of S.CPUE and nominal CPUE.

## INTRODUCTION

Species distribution models (SDMs) are the most common method of examining species habitat patterns through the use of oceanographic elements; they are also referred to as habitat models, ecological niche models, bioclimatic envelopes, and resource selection functions (*Zimmermann et al., 2010*; *Robinson et al., 2011*; *Beale & Lennon, 2012*; *Li & Wang, 2013*; *Tikhonov et al., 2020*). Habitat models use mathematical representations of the current species distribution to predict the future distribution by using an algorithm (*Azzellino et al., 2012*). Historically, the arithmetic mean and geometric mean models (*Xue et al., 2017*; *Li et al., 2016*) based on the habitat appropriateness index have been used.

Corresponding author
Ming An Lee,
malee@mail.ntou.edu.tw

However, the adoption of various machine learning models has been accelerated by technical advancement (*Zhang et al., 2021*). Machine learning (ML) is a topic of study focused on comprehending and developing "learning" techniques, or techniques that use data to improve performance on a certain set of tasks. Machine learning is the study of how computers use data to make predictions or judgments. Algorithms are used in fields such as medical; email filtering, speech recognition, agriculture, computer vision and most importantly in the field of biological oceanography. Machine learning is also known as predictive analytics when it comes to solving business problems. Different advanced models like regression models, including generalized linear models (GLMs) and generalized additive models (GAMs) as well as artificial intelligence models such as gradient boosting models (*Hossain et al., 2020*), artificial neural networks (*Ahmad, 2019*), classification and regression tree (CART) models (*Youssef et al., 2016*), and random forest models (*Reisinger et al., 2018*) are in use present days. Ensemble models (*Reisinger et al., 2021*; *Tabor & Koch, 2021*), which incorporate the forecasts of two or more habitat models (ensemble members), exhibit superior performance and robustness compared with single-algorithm models. The ensemble members may be identical or dissimilar habitat model types, and each can be trained with the same or different training sets (*Georgian, Anderson & Rowden, 2019*). Statistics like the mode or mean can be used to combine the ensemble members' predictions, and sophisticated techniques are employed to determine the trustworthiness of each ensemble member and the circumstances under which it can be relied upon (*Abrahms et al., 2019*). Many studies released in the 1990s have shown the extensive use of ensemble learning techniques such core bagging, boosting, and stacking. When a single model is insufficient, ensemble models can be used for prediction. Considering bias in the model helps lower the predictive error variance. The current study concentrated on ensemble modeling for the habitat of the black pomfret.

Black pomfret, or *Parastromateus niger* (Bloch, 1795), are small pelagic fish that live in the inshore waters of the Indian Ocean and western Pacific Ocean off the coasts of East Africa, southern Japan, and Australia (*Liu, 2008*). Black pomfret are found across the East China Sea and South China Sea, are prevalent in the Taiwan Strait, and play a significant role in lighting purse seine fisheries in Taiwan and in Fujian and Guangdong Provinces (*Lu & Youming, 1985*; *Lu, Quanshui & Youming, 1991*). Black pomfret production is vital to the coastal fisheries of Taiwan. However, the *P. niger* stock status in the Taiwan Strait (TS) recently collapsed (*Ju et al., 2020*), as evidenced by a significant drop in the annual capture in Taiwanese waters between 2002 and 2019 (*Fisheries Agency, Council of Agriculture, 2019*). Possible reasons behind this substantial decline in yield might be due to a combination of high demand, unregulated fishing methods, climate change, and the overfishing of *P. niger* (*Tao et al. (2012)*; *Ju et al. (2020)*), aimed/mentioned by the United Nations Sustainable Development Goal (SDG) 14 (*Ntona & Morgera, 2018*). To achieve SDG 14, overfishing must be controlled which is related to the unsustainable fish stock whereas underexploited or partially exploited levels are viable, the target of sustainable exploration.

Understanding *P. niger* habitat preferences and zones in-depth is essential for maintaining a species stock that is ecologically acceptable. In order to assist keep the
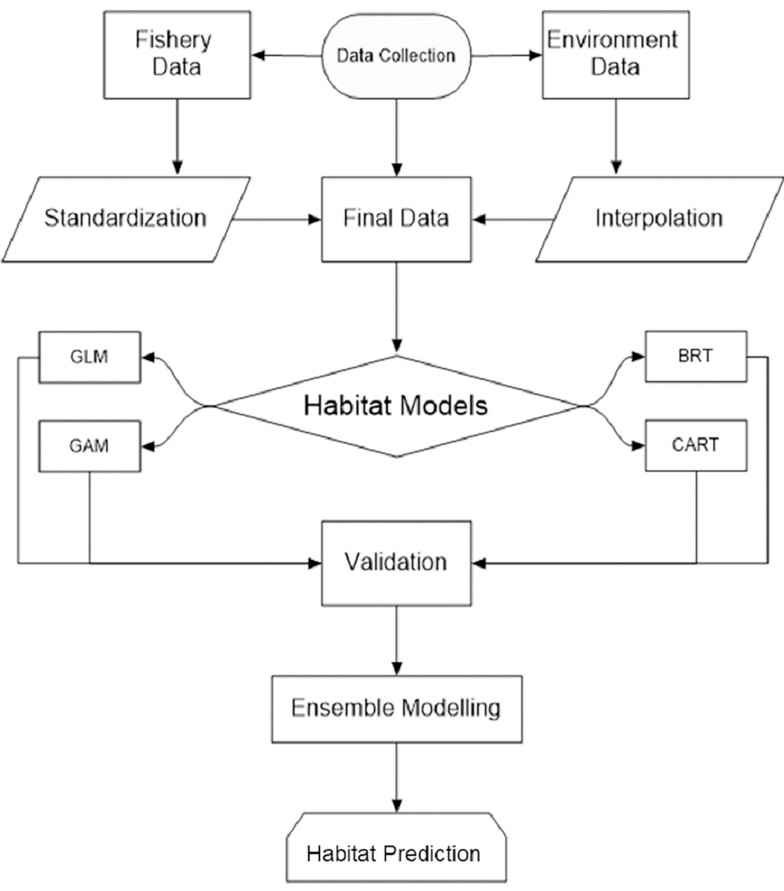

**Figure 1 Study flowchart.**

species stock within an ecologically acceptable range, we thus used habitat modeling to evaluate the effects of the maritime environment on the *P. niger* habitat (Fig. 1). (SDG 14.4). To stop overfishing of *P. niger*, a thorough study of its biological preferences and habitat regions may be helpful (SDG 14.6).

# MATERIALS AND METHODS

## Data collection

### Fisheries data

We collected information from Taiwanese fishing vessels (mostly coastal sea fishing, with gross register tonnage ranging from 0 to less than 250 tons) about *P. niger* fisheries from January 2014 to December 2019 from Taiwan's Fisheries Agency. The spatial coverage of the monthly fisheries data was 23°N–26°N and 118°E–120°E, with a resolution of 0.1°. The data provider did not specify whether the reported weights were dry or wet. Various fishing gear was used, and gears data with maximum catch contribution were only used in this study. The collected data included the year, month, latitude and longitude, catch in kilograms, effort in hours, total catch weight in one location, type of fishing gear used, and vessel identification number. Data on fishing depth and gear-soaking time were unavailable.

**Table 1 Surface oceanographic data sources and their descriptions.**

| Environmental data | Abbreviation | Unit | Spatial resolution | Temporal resolution |
|---|---|---|---|---|
| Sea surface temperature | SST | °C | 0.083° × 0.083° | Monthly |
| Sea surface salinity | SSS | psu | | |
| Mixed layer depth | MLD | m | | |
| Sea surface height above geoid | SSH | | | |
| Meridional velocity | U | ms$^{-1}$ | | |
| Zonal velocity | V | | | |
| Sea surface chlorophyll | SSC | mgm$^{-3}$ | 0.25° × 0.25° | Daily |

### Oceanographic data

Seven oceanographic characteristics (Table 1) were gathered from several sources for the current study: sea surface temperature (SST), sea surface salinity (SSS), mixed layer depth (MLD), sea surface chlorophyll (SSC) level, sea surface height (SSH), meridional velocity (U), and zonal velocity (V; Table 1). We calculated eddy kinetic energy (EKE) from U and V as follows: EKE = 0.5 (U$^2$ + V$^2$). The CMEMS eddy-resolving global ocean reanalysis product GLORYS12V1 (1/12° horizontal resolution with 50 vertical levels; https://resources.marine.copernicus.eu/product-detail/GLOBAL_MULTIYEAR_PHY_001_030/INFORMATION) was used to collect SST, SSS, MLD, SSH, U, and V data. Its processing level and coordinate reference system are L4 and W, respectively. In addition, we gathered SSC data using the CMEMS global ocean biogeochemical hindcast product FREEGLORYS2V4 (0.25° horizontal resolution, 75 vertical levels, daily temporal resolution; https://resources.marine.copernicus.eu/product-detail/GLOBAL_MULTIYEAR_BGC_001_029/INFORMATION), whose processing level and coordinate reference system are Level 4 and ETRS89 (EU-recommended frame of reference for geodata for Europe 1), respectively.

These data were originally gathered between January 2014 and December 2019 and covered the geographic range of 116°E–123°E and 21°N–26°N. These data were interpolated using MATBLAB (version 2019a) to a 0.1° spatial resolution to match the fisheries data. The SSC data were interpolated to a monthly temporal resolution using MATLAB in addition to the oceanographic and fisheries data.

### Fisheries data standardization

The relative abundance of *P. niger* was as assessed as the nominal catch per unit effort (N. CPUE) from a total of 55,852 observations as follows (*Dunn et al., 2000*; *Lauridsen et al., 2008*):

N.CPUE = (catch in kilograms)/(fishing effort in hours)

The use of the popular GLM standardization technique and resulting bias-filtered N. CPUE data (*Hazin et al., 2007*; *Hinton & Maunder, 2004*; *Tian et al., 2009*) helped to lessen the effects of spatial data, including latitude (lat) and longitude (long); temporal data (year and month); and interaction factors (*i.e.*, year*lat, lat*long, and year*long; *Mondal et al.,*

*2022*; *Mondal et al., 2021*; *Vayghan et al., 2020*; *Forrestal et al., 2019*; *Shono, 2004*). The key benefits of employing a GLM for standardization include the exponential distribution of response variables and the ability to employ categorical predictors. A stepwise GLM was created using the stats package in RStudio (version 3.6.0) using the aforementioned seven components (year, month, lat, long, year*lat, lat*long, and year*long). The family and procedure employed for GLM optimization were the Gaussian family and glm.fit, respectively. The GLM constructed for standardization was as follows:

$$\text{GLM}: \log(\text{N.CPUE}) = \text{year} + \text{month} + \text{lat} + \text{long} + \text{interactions},$$

where the interaction factors are year*lat, lat*long, and year*long.

## S.CPUE–oceanographic factor relationship

The correlations between the standardized catch per unit effort (S.CPUE) benchmark values and the aforementioned oceanographic factors were established to discern the preferred parameter ranges. We created suitability index (SI) curves for each oceanographic parameter using summed S with smoothing spline regression. The regression used S.CPUE as the dependent variable and all selected oceanographic elements as the explanatory variables. The SI curves were then normalized as follows using S.CPUE and the oceanographic variables:

$$\text{SI} = (Y - Y_{min})/(Y_{max} - Y_{min}),$$

where $Y_{max}$ and $Y_{min}$ are respectively the maximum and minimum number of observations of S.CPUE or oceanographic factors; thus, SI has a range between 0 and 1, where Y is a simulated or predicted value from Ymax to Ymin. An oceanographic factor range with a large SI value (>0.6) (*Mondal et al., 2021*; *Vayghan et al., 2020*; *Teng et al., 2021*) suggested a favorable range for S.CPUE.

## Single-algorithm habitat model development

The current study incorporated four single-algorithm models, namely a GLM, GAM, boosted regression tree (BRT) model, and CART model. Each modeling technique was optimized according to the established protocol. We developed one model for each modeling technique in RStudio and the six oceanographic factors (SST, SSS, MLD, SSH, SSC, and EKE), which were regarded as predictor variables; S.CPUE was the response variable.

We used the Gaussian family and the generalized cross-validation of the mgcv package to construct each GAM. We employed the Gaussian family and the glm.fit technique from the stats package to create each GLM. Each BRT model was built using the Gini approach and the Gaussian family from the gbm program; optimization included the use of 100 trees, seven interactions, and 0.65 bag fractions. The rpart package was used to build each CART model using the Gaussian family and the CP technique. The CART models were optimized with a CP value of 0.1 and minimum and maximum node counts of 1 and 6, respectively.
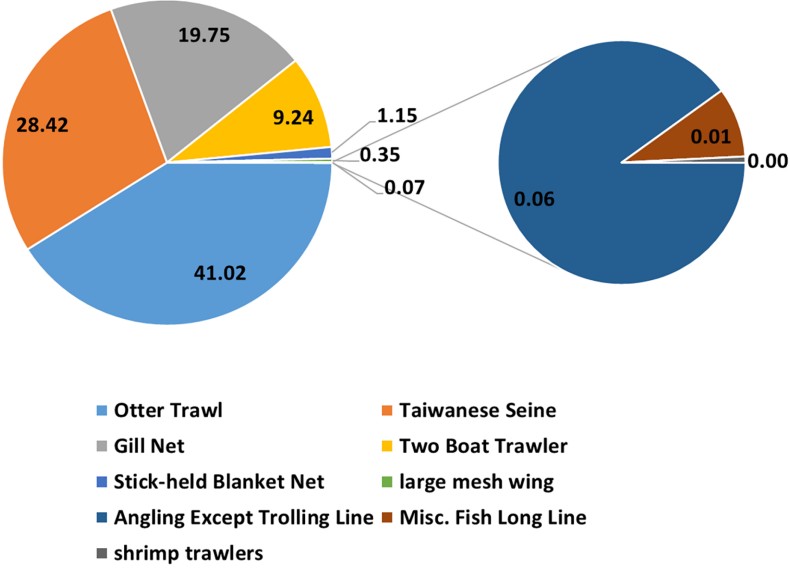

**Figure 2 Catches from fishing with different gear.**

## Validation of selected single-algorithm habitat models

The fisheries data set ($n$ = 55,852) was split into two portions using a random splitting technique performed by the RStudio caret package at a ratio of 70 ($n$(70) = 39,115) to 30 ($n$(30) = 16,737) to validate the single-algorithm models. For each single-algorithm model, three coefficients—namely the Pearson correlation coefficient (R), root-mean-square error (RMSE), and mean absolute error (MAE)—were computed for both portions of the data set. Little variation in the R, RMSE, and MAE values for the two data sets was considered indicative of a well-performing model with low bias.

## Ensemble habitat model development

We created an ensemble habitat model in the RStudio BIOMOD2 package (*Georgian, Anderson & Rowden, 2019*; *Alabia et al., 2016*; *Reisinger et al., 2021*; *Tabor & Koch, 2021*; *Abrahms et al., 2019*) to enhance the power to predict the *P. niger* habitat. A weighted mean ensemble model of the *P. niger* habitat was created after the performance of the single-algorithm models was assessed. If no discernible bias was detected on the basis of the R, RMSE, and MAE values for the two data sets for a single-algorithm model, the model was integrated into the ensemble model. Models exhibiting potential bias were excluded.

After the creation of the ensemble habitat model, MATLAB was used to visualize the monthly value predictions of the ensemble model along with the S.CPUE for each point in the study area.

## RESULTS

### Standardization of fisheries data

Over 88% of the catches were captured by otter trawl nets, gill nets, and Taiwanese seines (Fig. 2). Thus only these data were selected for the analysis. The full GLM (with all six

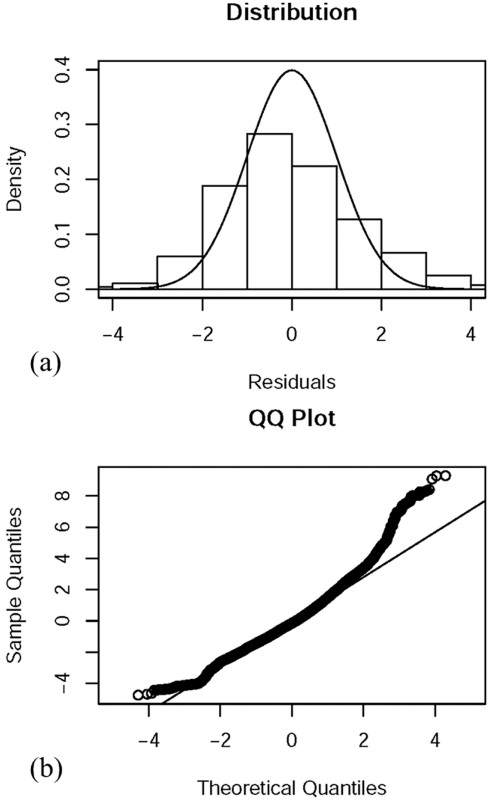

**Figure 3 Residual distributions and QQ graphs for the final GLM with predictor variables.**

factors) resulted in an explained deviance and adjusted $R^2$ of 18.135% and 0.181, respectively. The residual distribution and quantile–quantile (QQ) plots (Fig. 3) of the full GLM exhibited no significant fluctuation. Thus, the full GLM approach was used for the standardization of the *P. niger* fisheries data.

## S.CPUE–oceanographic factor relationships

The SI curves created for the S.CPUE of *P. niger* against the six oceanographic factors are illustrated in Fig. 4. When the SI value exceeded 0.6, the ideal SST, SSC level, SSS, MLD, SSH, and EKE ranges were 26.5–29.5 °C, 0.3–0.44 mg/m$^3$, 33.4–34.4 PSU, 10–14 m, 0.57–0.77 m, and 0.603–0.794 m$^2$/s$^2$, respectively. The total standardized CPUE of *P. niger* reached its maximum near the SST, SSC level, SSS, MLD, SSH, and EKE of 29.5 °C, 0.36 mg/m$^3$, 34.2 PSU, 12 m, 0.67 m, and 0.661–0.724 m$^2$/s$^2$, respectively. These results indicate the preferred oceanographic range of *P. niger* in the TS during the research period.

## Contributions of single oceanographic factors

Table 2 presents the performance of different oceanographic factors in the single-algorithm models. SSH was observed to be the most dominant oceanographic factor in all four single-algorithm models. The second most crucial oceanographic factor was EKE, which ranked second in all single-algorithm models except the GLM. SSC ranked third in all single-algorithm models except the GLM. SST, which ranked last in all the

Peer J

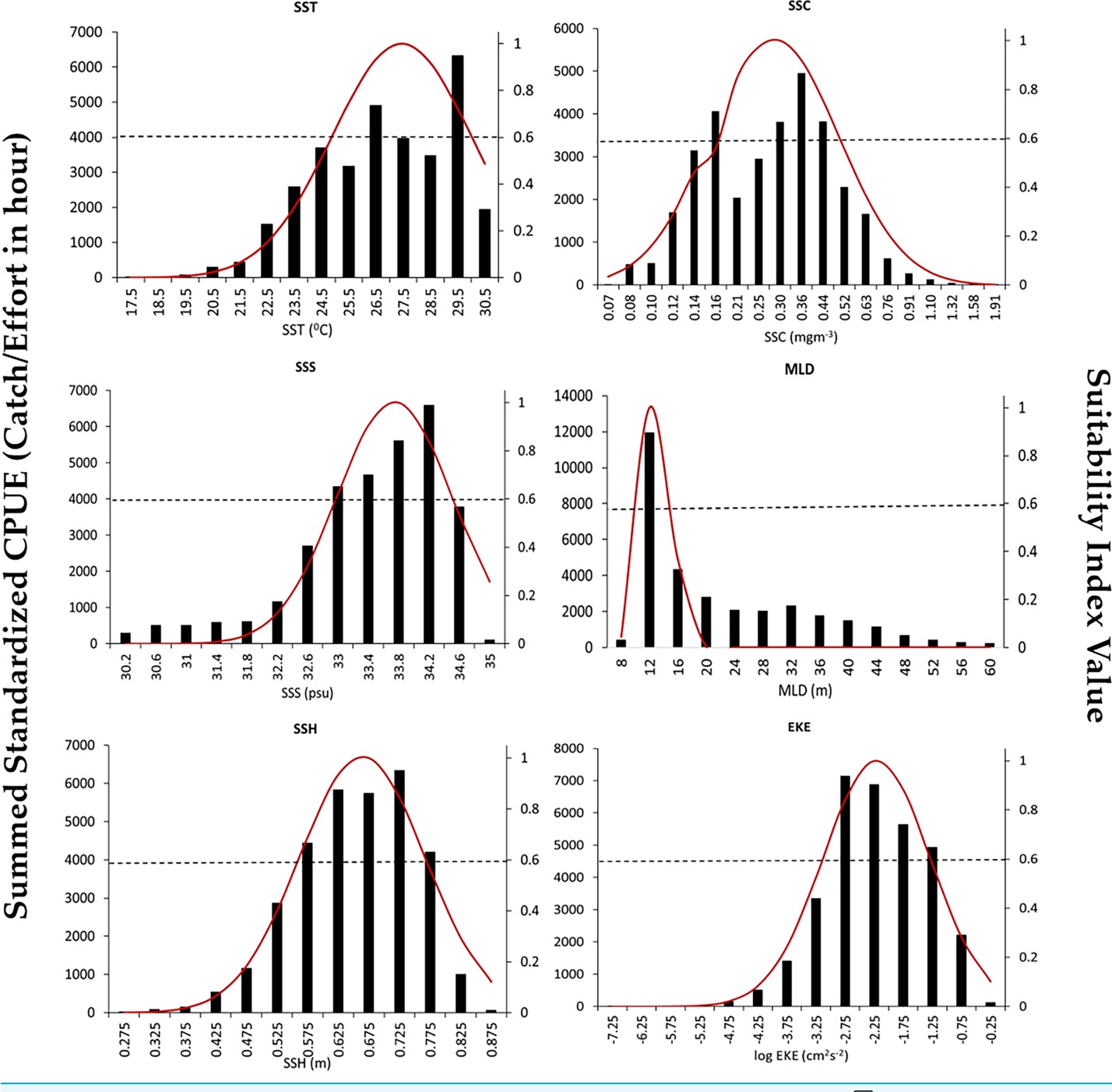

**Figure 4  Environmental ranges of *P. niger* with SI values from 2014 to 2019.**

models, was deemed the least critical parameter. SSS was ranked fifth in all models but the GLM, in which it was the least influential.

## Performance and validation of single-algorithm models

Table 3 presents the predictive performance of the full (with all oceanographic factors) GAM, GLM, BRT model, and CART model. The deviance explained by the full GAM,

**Table 2 The performance of different oceanographic factors in approaches based on GAMs, GLMs, BRTs, and CARTs.** Rank 1 parameters are shown in bold and represent the best parameter.

| Single-algorithm | SST | | SSC | | SSS | | MLD | | SSH | | EKE | |
|---|---|---|---|---|---|---|---|---|---|---|---|---|
| | Dev. | Rank | Dev. | Rank | Dev. | Rank | Dev. | Rank | Dev. | Rank | Dev. | Rank |
| GAM | 7.68 | 6 | 19.4 | 3 | 13.7 | 5 | 13.8 | 4 | 33.5 | *1* | 21.6 | 2 |
| GLM | 1.41 | 5 | 12.27 | 2 | 0.01 | 6 | 11.76 | 3 | 31.34 | *1* | 10.74 | 4 |
| BRT | 9.3 | 6 | 20.9 | 3 | 14.1 | 5 | 16.1 | 4 | 34.6 | *1* | 22.1 | 2 |
| RF | 8.9 | 6 | 18.5 | 3 | 10.9 | 5 | 15.1 | 4 | 33.2 | *1* | 20.6 | 2 |

**Table 3 The predictive performance of full models (with all oceanographic factors) of GAM, GLM, BRT, and CART.**

| Single-algorithm | $R^2$ | Dev. Exp. (%) |
|---|---|---|
| GAM | 0.543 | 54.1 |
| GLM | 0.415 | 41.52 |
| BRT | 0.525 | 52.7 |
| CART | 0.498 | 49.5 |

GLM, BRT model, and CART model was 54.1%, 41.52%, 52.7%, and 49.5%, respectively. The correlation values of the full GAM, GLM, BRT model, and CART model were 0.543, 0.415, 0.525, and 0.498, respectively. Figures 5A–5D depict the performance of the selected GAM, GLM, BRT model, and CART model, respectively.

Small differences in the coefficients (R, RMSE, and MAE) for the two randomly apportioned data sets (70:30) indicated no significant bias in any predictive (Table 4) models, and the predictions were mapped onto a 1° geographic grid.

### Ensemble habitat prediction

Because no discernible bias was detected on the basis of the R, RMSE, or MAE values for the 70% and 30% portions of the data, the produced ensemble was selected for final prediction (Table 5). Figure 6 presents the predicted CPUE (P.CPUE) and S.CPUE. A high annual S.CPUE was distributed primarily in the ranges of 119°E–121°E and 23°N–26°N, the coastal waters of Taiwan. Most S.CPUE values were >4 in these locations but <1 in the remaining study areas. P.CPUE displayed a pattern indicating expansion to 26°N. Both S. CPUE and P.CPUE were between 0.1 and 5.

## DISCUSSION

### Spatial distribution

High annual S.CPUE values were observed primarily in the ranges of 119°E–121°E and 23°N–26°N. The P.CPUE values displayed a comparable pattern, with extension to 26°N. This distribution pattern may result from various factors.

First, the Kuroshio Current and coastal currents have boosted species diversity and productivity in the waters near Taiwan (*Naimullah et al., 2020a*). The Kuroshio Branch

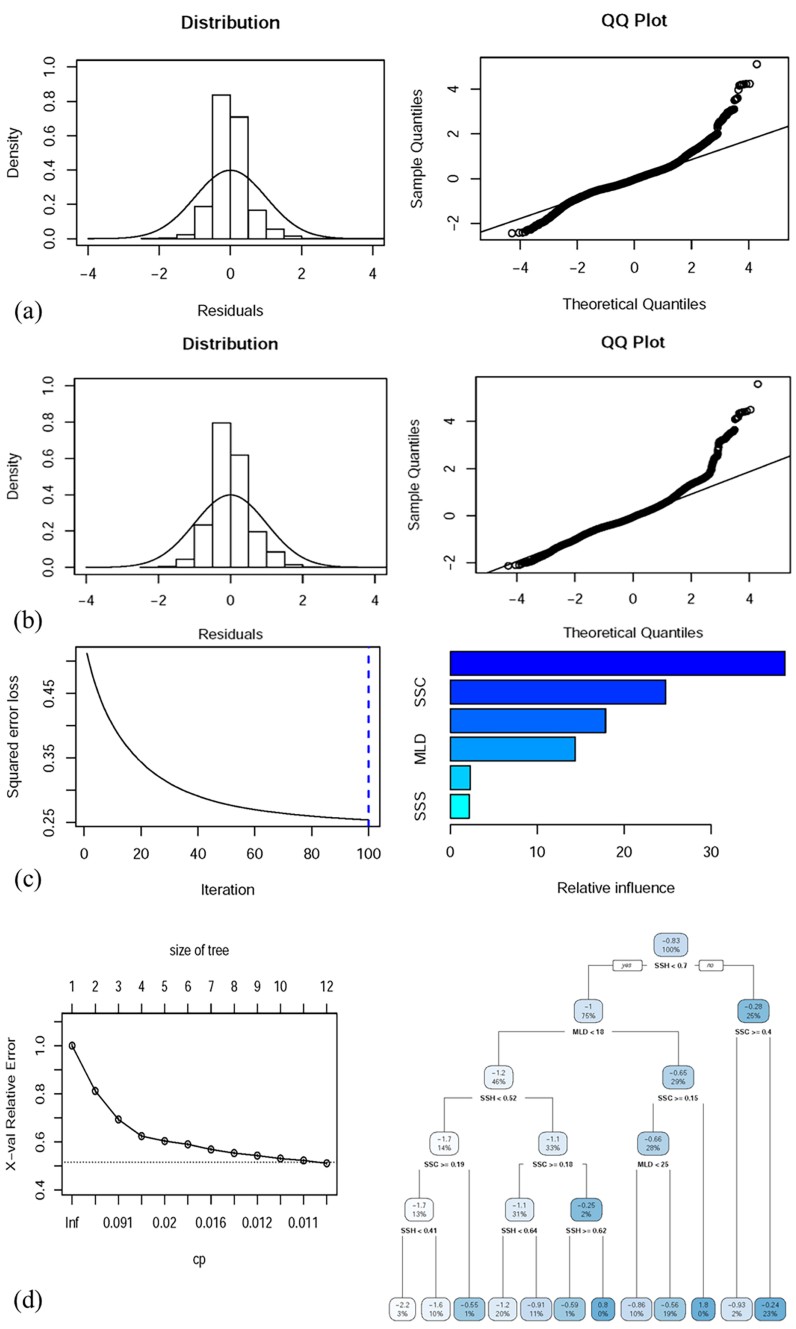

**Figure 5 Residual distributions and QQ plots for diagnostic analysis of the full (A) GAM and (B) GLM, both with predictor variables. Full (C) BRT and (D) CART model performance along with the decision trees of the final model.**

Current (KBC), China Coastal Current (CCC), and South China Sea (SCS) Current are three major currents that affect the TS, which is located in the tropical to subtropical western Pacific. These currents influence the fishing grounds and marine habitats of the East China Sea and SCS that border the TS to the north and south, respectively (*Naimullah et al., 2022*). The KBC provides a favorable environment for the diversified *P. niger* in the

**Table 4 Validation of selected single-algorithm models through random splitting.**

| Single- algorithm | 70% ($n$ = 39,115) | | | 30% ($n$ = 16,737) | | |
|---|---|---|---|---|---|---|
| | $R^2$ | RMSE | MAE | $R^2$ | RMSE | MAE |
| GAM | 0.479 | 1.452 | 1.344 | 0.482 | 1.448 | 1.343 |
| GLM | 0.361 | 1.454 | 1.331 | 0.345 | 1.455 | 1.332 |
| BRT | 0.528 | 0.503 | 0.373 | 0.527 | 0.502 | 0.371 |
| CART | 0.496 | 0.516 | 0.387 | 0.521 | 0.504 | 0.372 |

**Table 5 Ensemble habitat model's validation through random splitting.**

| 70% ($n$ = 39,115) | | | 30% ($n$ = 16,737) | | |
|---|---|---|---|---|---|
| $R^2$ | RMSE | MAE | $R^2$ | RMSE | MAE |
| 0.624 | 1.311 | 1.475 | 0.621 | 1.333 | 1.483 |

TS. The CCC offers a neritic water mass with low salinity and temperature but high nutrient content because of its connection to the rivers of the Chinese mainland (*Shiah et al., 2000*). Contrary to popular assumption, the KBC, which is derived from the Kuroshio Current, has high salinity and temperature and a nutrient level comparable to that of the CCC (*Chung, Jan & Liu, 2001*). These traits produce a water mass with physical characteristics distinct from those of the surrounding water. Properties such as temperature and salinity affect the distribution of *P. niger*. The trend indicated that South China Sea Water and Kuroshio Branch Water both invaded northward throughout the summer. The summer mean current on the eastern side can reach 90 cm/s in strength. The southwest monsoon is often lesser than 0.025 N/m$^2$ in the summer. Such insufficient wind force cannot propel a stream moving at 90 cm/s (*Jan et al., 2002*). As a result, rather than being driven by local winds, remote forcing with large-scale origin must drive a significant percentage of the circulation. The large-scale forcing is put up in such a way that it drives waters in the northern South China Sea to flow northward and enter the East China Sea through the Taiwan Strait. This might be the one possible reason behind the higher presence of black pomfret mainly on the southwestern coast of Taiwan during April to August. While the windward Kuroshio Branch Current on the eastern side is remotely driven in winter, the China Coastal Current on the western side is driven by the northeast monsoon and this can be the one possible reason behind the higher presence of black pomfret on the northeast coast during September to January.

Second, the KBC and CCC both contribute to upwelling. The bottom current in the TS flows upward from the continental slope, and the surface current is primarily driven by wind (*Naimullah et al., 2020b*). In addition, the eastern side of the TS receives occasional injections of water from the Kuroshio Current. The aforementioned upwelling forces nutrient-rich, typically chilly water to ascend to the surface. The nutrients "fertilize" the surface waters and thus support a high level of biological production (*Tang, Kawamura &*

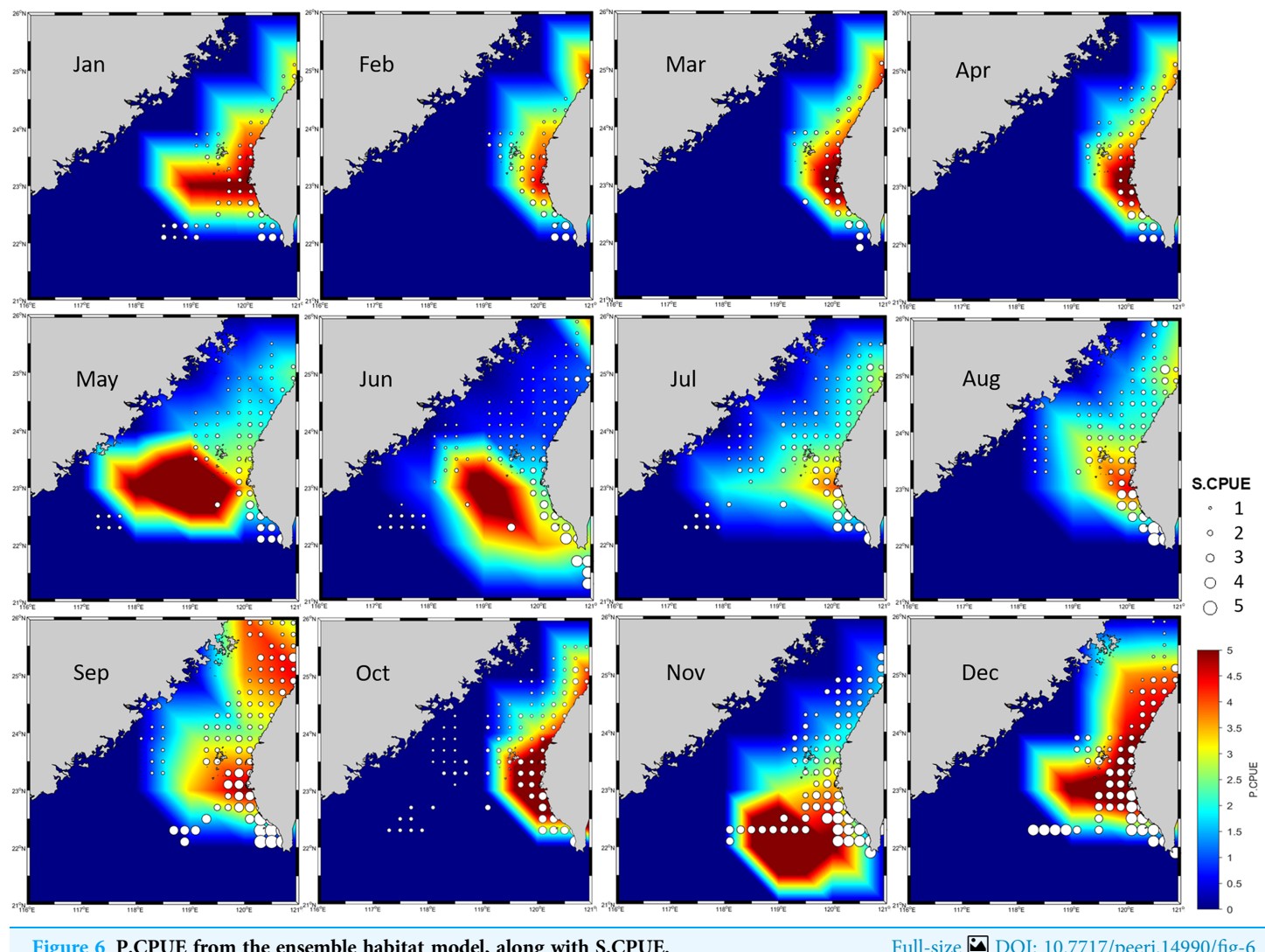

**Figure 6 P.CPUE from the ensemble habitat model, along with S.CPUE.**

*Guan, 2004*). Consequently, these fertilized TS zones may serve as ideal *P. niger* fishing locations. In addition to being a hydrological event, upwelling has a major effect on the ecology. The Taiwan Bank upwelling and Dongshan upwelling zones have good alignment with fishing grounds during the summer (*Tang et al., 2002*) on the west coast and might be the possible reason of higher presence of black pomfret on the south-west coast near Taiwan Bank during summer season.

Third, the seafloor of the TS is intricate. The seabed topography and capes influence tidal currents, which form counterclockwise eddies (*Lee et al., 2022*). From the topographic profile, it was noted that (*Lin, Juang & Tsay, 2000*) Taiwan Strait has a shelf-like topography from northwest to southwest part. In this, the higher tidal amplitude is present on the western part of Taiwan, which was shown as good fishing ground also in the present study. High chlorophyll concentrations outside the estuary are transferred by these tidal currents to the ocean current and attract secondary producers, including fish, crustaceans, and mollusks, and draw out *P. niger* for harvest.

## Habitat modeling approach for sustainable development

The Kuroshio Current and coastal currents near Taiwan contribute to the diversity and productivity of marine species. As a result, the prevalence of fleet-based fishing operations has grown substantially throughout Taiwan's waters over the past 40 years. The fishing gear used in this region includes purse seines, bottom and pelagic trawls, longlines, and gill and set nets (*Fisheries Agency, Council of Agriculture, 2019*). However, the trend of overfishing beginning in the 1950s caused catches to peak in 1980 and gradually decline afterward (*Chen, Lin & Chuang, 2018*; *Liao, Huang & Lu, 2019*). Despite frequent acknowledgment of the problematic state of coastal and offshore fisheries in Taiwanese waters (*Liu, 2013*; *Chen, 2006*; *Shao et al., 2011*), few fish species have been studied. Notably, the *P. niger* stocks in the waters close to Taiwan have drastically decreased (*Ju et al., 2020*).

The detailed information provided by habitat or spatial distribution modeling may assist in the sustainable management of *P. niger*. The pervasive nature of the measurement error inherent to models of species and habitat distribution may render such models unable to contribute to spatial economic optimization for sustainable planning. However, SDMs can potentially serve as heuristic tools for addressing oceanic environmental challenges. We emphasize the contextual application of such models.

Identification of fishing grounds that are underutilized or only partially utilized can be made easier using habitat models (*Rowden et al., 2017*). The predicted accuracy of single-algorithm models, however, can occasionally be impacted by data changes, leading to unduly optimistic or gloomy predictions. As a result, the current work used an ensemble modeling strategy. We merged and trained several single-algorithm models, often known as weak learners, to address the same issue. Weak learners ultimately produce ensemble models that are more accurate because, despite completing tasks poorly when working alone, they collaborate with other weak learners to become strong learners. The easy identification of fishing grounds crucially enhances fisheries revenue and reduces fishing effort, travel time, fuel consumption, and cost. However, the likelihood of such simplified identification of fishing grounds to result in overfishing highlights the relevance of the SDGs (*Mugagga & Nabaasa, 2016*). The adoption of SDG 14 has sparked discussion about ocean health and its importance to the future of the planet (*Ntona & Morgera, 2018*). In here the most important aspect is the conservation (*Virto, 2018*). Conservation measures can be taken in the overexploited areas and SDM can be used to identify initially the distribution zone of any particular species. Condition of these high or low catch zone can be examined through stock assessment to over or underexploited areas (*Kenny et al., 2018*). The SDG targets are intended address the major problems threatening ocean resources, such as overfishing and climate change (*Cormier & Elliott, 2017*; *Griggs et al., 2017*), but doing so requires emphases on the socioeconomic dimensions of ocean politics and the distinct positions of the least developed countries and small island states. The SDGs have garnered institutional acceptance since their adoption (*Friess et al., 2019*; *Sturesson, Weitz & Persson, 2018*). Understanding the habitat of *P. niger* in the TS may facilitate the sustainable management of the species. The primary aim of SDG 14.4 is

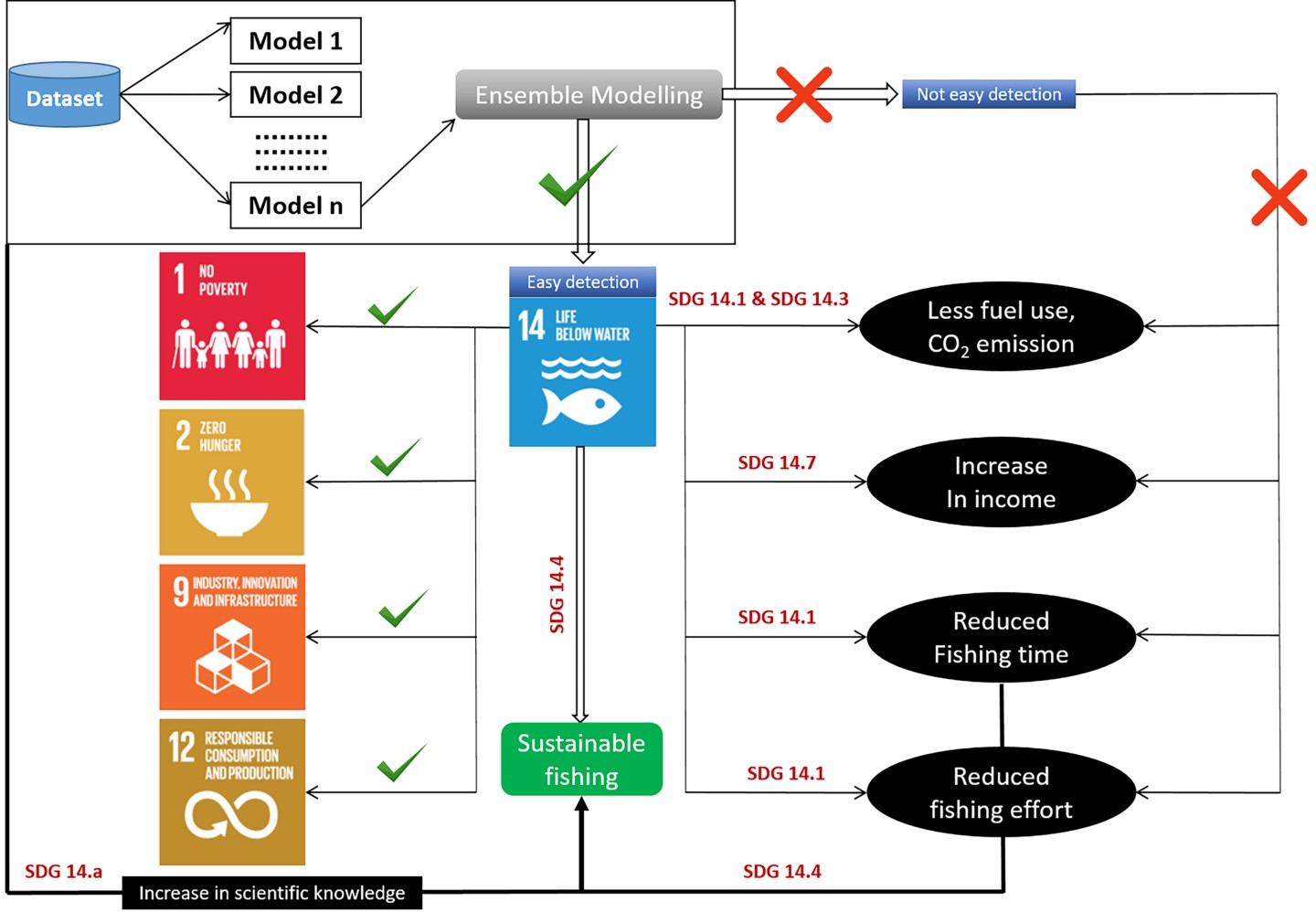

**Figure 7 Habitat modeling approach for sustainable development with SDG interactions.**

biologically sustainable fish stock levels. Habitat modeling can play a crucial role in achieving this goal by identifying the *P. niger* habitat in the TS. Additionally, SDG 14.5 focuses on conservation in coastal and marine areas. Highly exploited areas can be declared protected areas through temporary fishing prohibition to promote stock sustainability. SDG 14.6 calls for an end to overfishing subsidies. Subsidies for fishing vessels traveling to less-exploited areas should be discontinued to avoid overfishing. The sustainability of the oceans and their resources can also be promoted through the enhancement of scientific understanding, research, and the transfer of marine technology. Related policies should consider the Criteria and Guidelines of the Intergovernmental Oceanographic Commission (SDG 14.a), support small-scale fisheries (SDG 14.b), and implement and uphold international maritime law (SDG 14.c). The modeling of species distribution or habitats may constitute the initial stage in sustainability research (Fig. 7). According to *Ju et al. (2020)*, black pomfret stock in the Taiwan Strait is under collapsed condition and this result was supported by Taiwan Fishery Agency's year logbook. There was decreasing trends in black pomfret fisheries production and fisheries values from 2012

to 2021 with the value of 0.2 million tons and 20 billion NTD, respectively. These implies the importance of present study and we took habitat modelling as the first step for sustainable management of black pomfret fishery of Taiwan Strait.

Fisheries management organizations have developed and embraced ecosystem-based management techniques. The ability of oceans to meet the needs of their species is threatened (*Neumann, Ott & Kenchington, 2017*). As a result, many people may be forced to drastically reduce their demands on ocean ecosystems. The current study identified the detailed habitat preferences and zones of *P. niger* to further the maintenance of ecologically acceptable levels of species stock (SDG 14.4). A proper understanding of habitat preferences and zones can help to prevent the overfishing of *P. niger* (SDG 14.6). We plan to conduct future research on the predicted effects of climate change on *P. niger* through habitat-based modeling and to offer recommendations for sustainability.

## CONCLUSION

This study used a variety of oceanographic characteristics to pinpoint the geographic range of *P. niger* in the TS. Due to the GLM approach's superior performance to other models, we chose it for standardization. Near the SST, SSC level, SSS, MLD, SSH, and EKE of 29.5 °C, 0.36 mg/m$^3$, 34.2 PSU, 12 m, 0.67 m, and 0.661–0.724 m$^2$/s$^2$, respectively, the *P. niger* S.CPUE attained its highest value. According to the statistical analysis of our ensemble model, SST is the least important component and SSH and EKE are the key factors affecting the *P. niger* distribution. The largest yearly P.CPUE distribution followed by the largest annual S.CPUE distribution were found in the regions of 21°N–26°N and 119°E–121°E.

## ACKNOWLEDGEMENTS

We thank the anonymous reviewers and editors for their valuable comments and suggestions.

### Funding

This research was financed by the Council of Agriculture and the National Science and Technology Council of Taiwan. The Council of Agriculture and the National Science and Technology Council of Taiwan played a role in data collection. The funders had no role in study design, decision to publish, or the preparation of the manuscript.

### Grant Disclosures

The following grant information was disclosed by the authors:
Council of Agriculture and the National Science and Technology Council of Taiwan.

### Competing Interests

The authors declare that they have no competing interests.

## Author Contributions

- Sandipan Mondal conceived and designed the experiments, performed the experiments, analyzed the data, prepared figures and/or tables, authored or reviewed drafts of the article, and approved the final draft.
- Ming An Lee performed the experiments, prepared figures and/or tables, authored or reviewed drafts of the article, and approved the final draft.
- Yu-Kai Chen conceived and designed the experiments, authored or reviewed drafts of the article, and approved the final draft.
- Yi-Chen Wang analyzed the data, authored or reviewed drafts of the article, and approved the final draft.

## Data Availability

The raw data are available in the Supplemental File.

## Supplemental Information

Supplemental information for this article can be found online at http://dx.doi.org/10.7717/peerj.14990#supplemental-information.

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

# PeerJ

**Cormier R, Elliott M. 2017.** SMART marine goals, targets and management—is SDG 14 operational or aspirational, is 'Life Below Water' sinking or swimming? *Marine Pollution Bulletin* **123(1–2)**:28–33 DOI 10.1016/j.marpolbul.2017.07.060.

**Dunn A, Harley SJ, Doonan IJ, Bull B. 2000.** Calculation and interpretation of catch-per-unit effort (CPUE) indices. New Zealand Fisheries Assessment Report, 144.

**Fisheries Agency, Council of Agriculture. 2019.** Fisheries Annual Reports; Executive Yuan: Taipei, Taiwan.

**Forrestal FC, Schirripa M, Goodyear CP, Arrizabalaga H, Babcock EA, Coelho R, Ingram W, Lauretta M, Ortiz M, Sharma R, Walter J. 2019.** Testing robustness of CPUE standardization and inclusion of environmental variables with simulated longline catch datasets. *Fisheries Research* **210(4)**:1–13 DOI 10.1016/j.fishres.2018.09.025.

**Friess DA, Aung TT, Huxham M, Lovelock C, Mukherjee N, Sasmito S. 2019.** SDG 14: life below water-impacts on mangroves. *Sustainable Development Goals* **445**:445–481 DOI 10.1017/9781108765015.

**Georgian SE, Anderson OF, Rowden AA. 2019.** Ensemble habitat suitability modeling of vulnerable marine ecosystem indicator taxa to inform deep-sea fisheries management in the South Pacific Ocean. *Fisheries Research* **211(1)**:256–274 DOI 10.1016/j.fishres.2018.11.020.

**Griggs DJ, Nilsson M, Stevance A, McCollum D. 2017.** A guide to SDG interactions: from science to implementation. International Council for Science, Paris DOI 10.24948/2017.01.

**Hazin HG, Hazin F, Travassos P, Carvalho FC, Erzini K. 2007.** Standardization of swordfish CPUE series caught by Brazilian longliners in the Atlantic Ocean, by GLM, using the targeting strategy inferred by cluster analysis. *Collective Volume of Scientific Papers—ICCAT* **60(6)**:2039–2047.

**Hinton MG, Maunder MN. 2004.** Methods for standardizing CPUE and how to select among them. *Collective Volume of Scientific Papers—ICCAT* **56(1)**:169–177.

**Hossain MS, Sarker S, Sharifuzzaman SM, Chowdhury SR. 2020.** Primary productivity connects hilsa fishery in the Bay of Bengal. *Scientific Reports* **10(1)**:1–16 DOI 10.1038/s41598-020-62616-5.

**Jan S, Wang J, Chern CS, Chao SY. 2002.** Seasonal variation of the circulation in the Taiwan Strait. *Journal of Marine Systems* **35(3–4)**:249–268 DOI 10.1016/S0924-7963(02)00130-6.

**Ju P, Tian Y, Chen M, Yang S, Liu Y, Xing Q, Sun P. 2020.** Evaluating stock status of 16 commercial fish species in the coastal and offshore waters of Taiwan using the CMSY and BSM methods. *Frontiers in Marine Science* **7**:618 DOI 10.3389/fmars.2020.00618.

**Kenny AJ, Campbell N, Koen-Alonso M, Pepin P, Diz D. 2018.** Delivering sustainable fisheries through adoption of a risk-based framework as part of an ecosystem approach to fisheries management. *Marine Policy* **93(2)**:232–240 DOI 10.1016/j.marpol.2017.05.018.

**Lauridsen TL, Landkildehus F, Jeppesen E, Jørgensen TB, Søndergaard M. 2008.** A comparison of methods for calculating catch per unit effort (CPUE) of gill net catches in lakes. *Fisheries Research* **93(1–2)**:204–211 DOI 10.1016/j.fishres.2008.04.007.

**Lee HJ, Lee MA, Ho CY, Hsu PC, Wang YC. 2022.** Crescent-shaped low-temperature distribution along the convex topography in the southeastern edge of Taiwan Bank in summer. *Journal of Physical Oceanography* **52(11)**:2705–2723.

**Li G, Cao J, Zou X, Chen X, Runnebaum J. 2016.** Modeling habitat suitability index for Chilean jack mackerel (*Trachurus murphyi*) in the South East Pacific. *Fisheries Research* **178**:47–60 DOI 10.1016/j.fishres.2015.11.012.

**Li X, Wang Y. 2013.** Applying various algorithms for species distribution modelling. *Integrative Zoology* **8(2)**:124–135 DOI 10.1111/1749-4877.12000.

Liao CP, Huang HW, Lu HJ. 2019. Fishermen's perceptions of coastal fisheries management regulations: key factors to rebuilding coastal fishery resources in Taiwan. *Ocean & Coastal Management* **172**:1–13 DOI 10.1016/j.ocecoaman.2019.01.015.

Lin M-C, Juang W-J, Tsay T-K. 2000. Application of the mild-slope equation to tidal computations in the Taiwan Strait. *Journal of Oceanography* **56**:625–642.

Liu RY. 2008. *Checklist of marine biota of China Seas*. Beijing: Science Press, 1267.

Liu WH. 2013. Managing the offshore and coastal fisheries in Taiwan to achieve sustainable development using policy indicators. *Marine Policy* **39(3)**:162–171 DOI 10.1016/j.marpol.2012.11.001.

Lu Z, Quanshui D, Youming Y. 1991. Growth and mortality of *Auxis thazard* in the Taiwan Strait and its adjacent sea. *Journal of Fisheries of China* **15(3)**:228–235.

Lu Z, Youming Y. 1985. Age and growth of *Parastromateus niger* in Western Taiwan Strait. *Journal of Fujian Fisheries* **3**:7–13.

Mondal S, Vayghan AH, Lee MA, Wang YC, Semedi B. 2021. Habitat suitability modeling for the feeding ground of immature albacore in the southern Indian Ocean using satellite-derived sea surface temperature and chlorophyll data. *Remote Sensing* **13(14)**:2669 DOI 10.3390/rs13142669.

Mondal S, Wang YC, Lee MA, Weng JS, Mondal BK. 2022. Ensemble three-dimensional habitat modeling of Indian Ocean immature albacore tuna (*Thunnus alalunga*) using remote sensing data. *Remote Sensing* **14(20)**:5278 DOI 10.3390/rs14205278.

Mugagga F, Nabaasa BB. 2016. The centrality of water resources to the realization of sustainable development goals (SDG). A review of potentials and constraints on the African continent. *International Soil and Water Conservation Research* **4(3)**:215–223 DOI 10.1016/j.iswcr.2016.05.004.

Naimullah M, Lan KW, Liang YR, Hsiao PY, Chiu TC, Liao CH. 2020a. Distribution and habitat characteristics of three important commercial swimming crab (Crustacea: Decapoda: Portunidae) related with the environmental factors in Taiwan Strait. *Available at http://scholars.ntou.edu.tw/handle/123456789/15802*.

Naimullah M, Lan KW, Liao CH, Hsiao PY, Liang YR, Chiu TC. 2020b. Association of environmental factors in the Taiwan Strait with distributions and habitat characteristics of three swimming crabs. *Remote Sensing* **12(14)**:2231 DOI 10.3390/rs12142231.

Naimullah M, Lee WY, Wu YL, Chen YK, Huang YC, Liao CH, Lan KW. 2022. Effect of soaking time on targets and bycatch species catch rates in fish and crab trap fishery in the southern East China Sea. *Fisheries Research* **250**:106258 DOI 10.1016/j.fishres.2022.106258.

Neumann B, Ott K, Kenchington R. 2017. Strong sustainability in coastal areas: a conceptual interpretation of SDG 14. *Sustainability Science* **12(6)**:1019–1035 DOI 10.1007/s11625-017-0472-y.

Ntona M, Morgera E. 2018. Connecting SDG 14 with the other sustainable development goals through marine spatial planning. *Marine Policy* **93(1)**:214–222 DOI 10.1016/j.marpol.2017.06.020.

Reisinger RR, Friedlaender AS, Zerbini AN, Palacios DM, Andrews-Goff V, Dalla Rosa L, Double M, Findlay K, Garrigue C, How J, Jenner C. 2021. Combining regional habitat selection models for large-scale prediction: circumpolar habitat selection of Southern Ocean humpback whales. *Remote Sensing* **13(11)**:2074 DOI 10.3390/rs13112074.

Reisinger RR, Raymond B, Hindell MA, Bester MN, Crawford RJ, Davies D, de Bruyn PN, Dilley BJ, Kirkman SP, Makhado AB, Ryan PG. 2018. Habitat modelling of tracking data from

multiple marine predators identifies important areas in the Southern Indian Ocean. *Diversity and Distributions* **24(4)**:535–550 DOI 10.1111/ddi.12702.

**Robinson LM, Elith J, Hobday AJ, Pearson RG, Kendall BE, Possingham HP, Richardson AJ. 2011.** Pushing the limits in marine species distribution modelling: lessons from the land present challenges and opportunities. *Global Ecology and Biogeography* **20(6)**:789–802 DOI 10.1111/j.1466-8238.2010.00636.x.

**Rowden AA, Anderson OF, Georgian SE, Bowden DA, Clark MR, Pallentin A, Miller A. 2017.** High-resolution habitat suitability models for the conservation and management of vulnerable marine ecosystems on the Louisville Seamount Chain, South Pacific Ocean. *Frontiers in Marine Science* **4**:335 DOI 10.3389/fmars.2017.00335.

**Shao KT, Soong KY, Lin CW, Wu SP, Chan TY, Chang JS. 2011.** Investigation and planning of fishery resources conservation zones and rare species. Fishery Agency, Taipei (in Chinese). *Available at* https://en.fa.gov.tw/view.php?theme=Fishery_Conservation_Zone&subtheme=&id=1.

**Shiah FK, Chung SW, Kao SJ, Gong GC, Liu KK. 2000.** Biological and hydrographical responses to tropical cyclones (typhoons) in the continental shelf of the Taiwan Strait. *Continental Shelf Research* **20(15)**:2029–2044 DOI 10.1016/S0278-4343(00)00055-8.

**Shono H. 2004.** A review of some statistical approaches used for CPUE standardization. Bulletin of the Japanese Society of Fisheries Oceanography (Japan). *Available at* https://agris.fao.org/agris-search/search.do?recordID=JP2005001680.

**Sturesson A, Weitz N, Persson Å. 2018.** SDG 14: life below water. A review of research needs. Technical Annex to the Formas Report Forskning för Agenda, 2030.

**Tabor JA, Koch JB. 2021.** Ensemble models predict invasive bee habitat suitability will expand under future climate scenarios in Hawai'i. *Insects* **12(5)**:443 DOI 10.3390/insects12050443.

**Tang D, Kester DR, Ni IH, Kawamura H, Hong H. 2002.** Upwelling in the Taiwan Strait during the summer monsoon detected by satellite and shipboard measurements. *Remote Sensing of Environment* **83(3)**:457–471.

**Tang DL, Kawamura H, Guan L. 2004.** Long-time observation of annual variation of Taiwan Strait upwelling in summer season. *Advances in Space Research* **33(3)**:307–312 DOI 10.1016/S0273-1177(03)00477-0.

**Tao Y, Mingru C, Jianguo D, Zhenbin L, Shengyun Y. 2012.** Age and growth changes and population dynamics of the black pomfret (*Parastromateus niger*) and the frigate tuna (*Auxis thazard* thazard), in the Taiwan Strait. *Latin American Journal of Aquatic Research* **40(3)**:649–656 DOI 10.3856/vol40-issue3-fulltext-13.

**Teng SY, Su NJ, Lee MA, Lan KW, Chang Y, Weng JS, Wang YC, Sihombing RI, Vayghan AH. 2021.** Modeling the habitat distribution of Acanthopagrus schlegelii in the coastal waters of the Eastern Taiwan strait using MAXENT with fishery and remote sensing data. *Journal of Marine Science and Engineering* **9(12)**:1442.

**Tian S, Chen X, Chen Y, Xu L, Dai X. 2009.** Standardizing CPUE of *Ommastrephes bartramii* for Chinese squid-jigging fishery in Northwest Pacific Ocean. *Chinese Journal of Oceanology and Limnology* **27(4)**:729–739 DOI 10.1007/s00343-009-9199-7.

**Tikhonov G, Opedal ØH, Abrego N, Lehikoinen A, de Jonge MM, Oksanen J, Ovaskainen O. 2020.** Joint species distribution modelling with the R-package Hmsc. *Methods in Ecology and Evolution* **11(3)**:442–447 DOI 10.1111/2041-210X.13345.

**Vayghan AH, Lee MA, Weng JS, Mondal S, Lin CT, Wang YC. 2020.** Multisatellite-based feeding habitat suitability modeling of albacore tuna in the southern Atlantic ocean. *Remote Sensing* **12(16)**:2515 DOI 10.3390/rs12162515.

**Virto LR. 2018.** A preliminary assessment of the indicators for sustainable development goal (SDG) 14 conserve and sustainably use the oceans, seas and marine resources for sustainable development. *Marine Policy* **98(5891)**:47–57 DOI 10.1016/j.marpol.2018.08.036.

**Xue Y, Guan L, Tanaka K, Li Z, Chen Y, Ren Y. 2017.** Evaluating effects of rescaling and weighting data on habitat suitability modeling. *Fisheries Research* **188**:84–94 DOI 10.1016/j.fishres.2016.12.001.

**Youssef AM, Pourghasemi HR, Pourtaghi ZS, Al-Katheeri MM. 2016.** Landslide susceptibility mapping using random forest, boosted regression tree, classification and regression tree, and general linear models and comparison of their performance at Wadi Tayyah Basin, Asir Region, Saudi Arabia. *Landslides* **13(5)**:839–856 DOI 10.1007/s10346-015-0614-1.

**Zhang T, Song L, Yuan H, Song B, Ebango Ngando N. 2021.** A comparative study on habitat models for adult bigeye tuna in the Indian Ocean based on gridded tuna longline fishery data. *Fisheries Oceanography* **30(5)**:584–607 DOI 10.1111/fog.12539.

**Zimmermann NE, Edwards TC Jr, Graham CH, Pearman PB, Svenning JC. 2010.** New trends in species distribution modelling. *Ecography* **33(6)**:985–989 DOI 10.1111/j.1600-0587.2010.06953.x.