# Peer review of "Ensemble modeling of black pomfret (Parastromateus niger) habitat in the Taiwan Strait based on oceanographic variables"

_PeerJ, doi:10.7717/peerj.14990_

## Round 0.1 · original submission · Minor Revisions

As you can see, two reviewers provided detailed revisions to your manuscript. They appreciated the work and the organisation and clarity of the text. They highlighted a number of specific issues, providing detailed information and constructive suggestions that should help you in the revision.

Reviewer 1 ·

Basic reporting

The manuscript titled "Ensemble modeling of black Pomfret (Parastromateus niger) habitat in the Taiwan Strait based on oceanographic variables" used the data from Taiwan's Fisheries Agency to illustrate the geographic distribution of Parastromateus niger in the Taiwan Strait. Moreover, generalized linear models are performed throughout the distribution models. This paper addresses an important marine biological problem and is aligned with the journal's aims and scope. I read this manuscript with interest. I think the article is suitable to be accepted in this journal. However, important issues remain to be addressed, which means a minor revision is required before the manuscript can be considered for publication. Detailed and point-by-point comments are provided below for your attention:

(I) Please rewrite the abstract, as the abstract does not show the research's descriptive results.
(ii) The authors sometimes describe the standard and appropriate terminologies. However, It's suggestive of illustrating and describing the language shortly.
(iii) The manuscript follows a good structure, and the figures and tables are mostly excellent. I suggest modifying and re-structure the conclusions. It's suggestive of blending the text's numbered conclusions.
(iv) The structure of illustrations presented in the "Discussion" section needs a revisit and a more detailed approach to delivering the impacted result.
(v) The authors used many references in the article. However, significantly less has been mentioned while reviewing the related work and not mentioning recent advancements in the Introduction. When discussing the related literature, I think you must also say the methods and procedures a little. This can further justify the need for the research presented in this paper.
(vi) Further, the last paragraph of the "Introduction" section should describe the existing knowledge gap and then tell the study aims and objectives and the novelty/new information generated from this paper.
(vii) It's out of curiosity whether any other physical model fits this situation.
(viii) Numerical descriptions are largely fine; reading through your method. A minor point: it's better to mention which function or codes are adopted (for the Matlab case and other places where suits).
(ix) The analysis of the results is satisfactory and of good quality. However, I think more discussion can be added to the figures and tables.

Experimental design

No comment

Validity of the findings

No comment

Additional comments

I will recommend the authors modify the style set of writing: PSU, not psu, m^2/s^{-2} etc.

Reviewer 2 ·

Excellent Review

This review has been rated excellent by staff (in the top 15% of reviews)
EDITOR COMMENT
The reviewer provided a very detailed revision of the manuscript, giving very clear and insightful comments on specific comments without losing the view of the big picture. All the comments were presented constructively and - in most cases- offered a possible solution to the authors on how to deal with the specific issues.

Basic reporting

This manuscript focused on Black Pomfret Parastrometeus niger species distribution modeling in the Taiwan Strait based on the ensemble habitat modeling of four single-algorithm approaches, including GLM, GAM, BRT, and CART. This study aims to improve the prediction of the target species' habitat and keep an acceptable fishing stock within an ecologically sustainable range (SDG 14.4).
In general, the manuscript writing is quite straightforward and makes the reader easy to follow the science behind it. As the title of this research, it already achieved the aims proposed with several associated and necessary illustrations. However, this manuscript should be strengthened with the following suggestions:
1. Lines 65-66: The authors mentioned “This substantial decline in yield is due to a combination of high demand, unregulated fishing methods, climate change, eutrophication, and the overfishing of P. niger” without evidence for this statement. Unless no citation or evidence could support this statement, it is relatively difficult to convince the reader to believe.
2. Lines 67-76: Sustainable Development Goals were mentioned with lots of citations, yet they should be expressed more clearly and tried to connect between SDGs and the authors’ ideas. Don’t mention too many unrelated statements that are difficult for the reader to understand what the authors explained.
3. Line 89: In this study, the authors collected “small-scale” fisheries data. Here, the “small-scale” definition was “mostly offshore sea fishing, 100 gross register tonnage and 24m in length”. Smith and Basurto (2019) provided the most popular feature used to determine the small-scale fishery were the type of fishing gear (58%), and vessels (51%; length, type of material, tonnage) whereas distance from shore was utilized moderately. Previous recent studies pointed out that the small-scale vessel size was below 12m (eg: Natale et al., 2015; Zhao and Jia, 2020) or the maximum tonnage of fishing boats was 5 tons (eg: Halim et al., 2019) based on their own countries’ regulation or the law. In Taiwan, the current Fisheries Act has yet to prescribe and determine small-scale fisheries; however, several scholars defined small-scale fishing vessels were less than 20 tons in capacity (Hsu et al., 2019; Lee et al., 2021). Therefore, what is the basis for defining the small-scale fisheries in this manuscript?
4. Line 89: The author defined the maximum tonnage and size of small-scale vessels were 100 tons and 24m in length; however, the collected raw data of this study consisted of CT5 and CT6, whose tonnage was greater than 100 tons (referred from Marine Bureau, Kaohsiung City government: https://kcmb.kcg.gov.tw/cp.aspx?n=7988CDDC702EB309). Furthermore, in the fisheries data, no information related to the fishing vessel size was mentioned. As the above issues, the authors should explain clearly.
5. Line 209: In the manuscript, the authors considered the ideal environmental variables were defined when the SI value should exceed 0.6. The question is why the SI value is greater than 0.6, not 0.5? How to explain this? Could the authors provide any evidence or previous studies that support this statement?
6. Lines 263-270: The authors discussed the role of the Kuroshio and other coastal currents near Taiwan, affecting the diversity and productivity of marine species in the waters off Taiwan. However, the topic of this research focused on the Taiwan Strait. Whether these ideas are too larger for the study area? In this part, the reader suggests the authors should try to link and focus on the Taiwan Strait, the study area by selecting the related information.
7. Figure 6: The topic of this manuscript focuses on the P. niger habitat modeling in the Taiwan Strait, yet figure 6 illustrates the predicted CPUE distribution in the waters off Taiwan including non-Taiwan Strait areas. It should be revised carefully and the unrelated marine sites should be removed.
8. Line 271: The authors commented that “the P. niger stocks in the waters close to Taiwan have drastically decreased” and cited the publication of Ju et al. (2020), collecting the fisheries data from Taiwan, Kinmen, and Matsu areas between 1949 and 2019. Discussion of the publication of Ju et al. (2020) mentioned that the decreasing trend in P. niger stock in the Taiwan Strait was demonstrated by Tao et al. (2012); therefore, it is believed that the authors should find the publication of Tao et al. (2012), not Ju et al. (2019).
9. In 4.1, the authors discussed the spatial distribution of P. niger in the Taiwan Strait with the effects of three major oceanic currents in the Taiwan Strait: The Kuroshio Branch Current, the China Coastal Current, and the South China Sea warm current. However, Jan et al. (2002) pointed out that the activities of the Northeastern monsoon and topography were the important factors controlling the northward intrusion of the Kuroshio Branch and South China Sea warm currents. Furthermore, Kuo et al. (2018) emphasized the considerable effects of topography on the circulation in the Taiwan Strait, especially Chang-Yun Rise. Also, Tang et al. (2002) identified five upwelling zones in the Taiwan Strait and summarized temporal observations of those upwelling with their impacts on the season of fishing grounds. From the reader’s point of view, the authors should explain the seasonality of three currents and the possible impacts on the spatial distribution through time (see Fig. 6 in the manuscript).
10. Figure 6 illustrated the predicted CPUE of Black Pomfret based on the ensemble habitat modeling. Many previous studies also emphasized the usefulness of the ensemble approaches for identifying species distribution forecasting with high accuracy (eg: Buisson et al., 2010; Grenouillet et al., 2011; Lin et al., 2018; Hysen et al., 2022); however, the results of the Black Pomfret distribution prediction (see Figure 6) remain that there still have several high predicted CPUE point beyond the red area (high predicted CPUE regions). What is the uncertainty of this ensemble habitat modeling?
11. Regarding the discussion on Habitat Modelling Approach for Sustainable Development, the authors believed that habitat modeling could achieve the Black Pomfret habitat identification, contributing to promoting stock sustainability and avoiding overfishing. However, from a personal point of view, the authors should provide more evidence for this statement and explain more details on related fisheries management policy.
12. In terms of the importance of this study, the reader agrees with the authors’ ideas partly, the connection between the role of ensemble habitat modeling with SDGs. As shown in Figure 7, the ensemble modeling could contribute to SDG 14, having a relationship with SDG 1, SDG 2, SDG 12, and SDG 14. Generally, the interactions between SDGs are believed to be possible, however, this manuscript only focused on one species – Black Pomfret Parastromateus niger. Based on the fisheries statistic yearbook in 2022 (downloaded via the website of Taiwan Fishery Agency https://www.fa.gov.tw/view.php?theme=FS_AR&subtheme=&id=21), there were decreasing trends in fisheries production and fisheries values from 2012 to 2021 with the value of 0.2 million tons and 20 billion NTD, respectively. Furthermore, Black Pomfret excludes from the list of 15 major species in Taiwan (fisheries statistic yearbook, 2022). Therefore, it should be careful and give a more powerful argument when explaining the connection between Black Pomfret habitat modeling and other SDGs (SDG 1, SDG 2, SDG 9, and SDG 12).

I look forward to receiving your revised manuscript.

References:
1. Buisson L, Thuiller W, Casajus N, Lek S, Grenouillet G (2010) Uncertainty in ensemble forecasting of species distribution. Global Change Biology, 16: 1145-1157.
2. Grenouillet G, Buisson L, Casajus N, Lek S (2011) Ensemble modelling of species distribution: the effects of geographical and environmental ranges. Ecography, 34: 9-17.
3. Halim A, Wiryawan B, Loneragan NR, Hordy A, Sondita MFA, White AT, Sonny Koeshendrajana S, Ruchimat T, Pomeroy RS, Yuni C (2019) Developing a functional definition of small-scale fisheries in support of marine capture fisheries management in Indonesia. Marine Policy, 100: 238-248.
4. Hsu WY, Wang SY, Hong WS, Hu RH, Yu CJ, Tasi HY (2019) Portable Fisheries Assistant Systems for Small Scale Fisheries Management. 2019 IEEE Eurasia Conference on IOT, Communication and Engineering, 10-13.
5. Hysen L, Nayeiri D, Cushman S, Wan HY (2020) Background sampling for multi-scale ensemble habitat selection modelling: Does the number of points matter? Ecological Informatics, 72: 101914.
6. Jan S, Wang J, Chern CS, Shao SY (2002) Seasonal variation of the circulation in the Taiwan Strait. Journal of Marine systems, 35: 249-268.
7. Kuo Y., Chan J, Wang YC, Shen YL, Chang Y, Lee MA (2018) Long-term observation on sea-surface temperature variability in the Taiwan Strait during the northeast monsoon season. International journal of Remote Sensing, 39(13): pp 4330-4342.
8. Lee YJ, Su NJ, Lee HT, Hsu WWY, Liao CH (2021) Application of Métier-Based Approaches for Spatial Planning and Management: A Case Study on a Mixed Trawl Fishery in Taiwan. Journal of Marine Science and Engineering, 9, 480.
9. Lin YP, Lin WC, Wu WY (2018) Uncertainty in Various Habitat Suitability Models and Its Impact on Habitat Suitability Estimates for Fish. Water, 7: 4088-4107.
10. Natale F, Carvalho N, Paulrud A (2015) Defining small-scale fisheries in the EU on the basis of their operational range of activity: The Swedish fleet as a case study. Fisheries Research, 164: 286-292.
11. Smith H, Basurto X (2019) Defining small-scale fisheries and examing the role of science in shaping perceptions of who and what counts: A systematic review. Fronties in Marine Science, 6: 236.
12. Zhao X, Jia P (2020) Towards sustainable small-scale fisheries in China: A case study of Hainan. Marine Policy, 121: 103935.

Experimental design

The article is within the aims and scope of the journal with a well-defined research question, but the minor issues of this manuscript should be revised as the following:
1. The arrangement of citations in the text should be double-checked. Should citation in the text follow the increased time order like Lines 36-37 (it should be “(Zimmerman et al., 2010; Robinson et al., 2011…)”).
2. Several mistakes should be corrected, like a unit of SSC (mg/m3 not mg/m-3).
3. Line 93-94: “Various fishing gear was used, but over 88% of the catches were captured by otter 94 trawl nets, gill nets, and Taiwanese seines (Figure 2)” this statement should be in the results or discussion.
4. Line 111, 116: What are L4, W, and ETRS 89?
5. Line 157-158: “Ymax and Ymin are respectively….” should be corrected as “Ymax and Ymin are…., respectively”.
6. Line 272-274: The comment “Fisheries management…. ecosystem services (Kenny et al., 2018)” should be moved to part of 4.2 or 4.3.
7. Line 274-275: The statement “This study thoroughly examined… P. niger fisheries” should be moved to the conclusion.

Validity of the findings

No comment

Additional comments

Please ensure that all review comments are addressed and inserted into the revised manuscript.

Annotated reviews are not available for download in order to protect the identity of reviewers who chose to remain anonymous.

---

## Round 0.2 · Minor Revisions

As you can see, one reviewer still raises some specific comments on your manuscript. Please consider them carefully while editing the manuscript and explain how each suggestion was adopted or rejected.

Reviewer 2 ·

Basic reporting

Thank you for taking the time to address the reviewer comments thoroughly, yet the manuscript still has many unresolved issues as the following:
1. Line 42-54: This part of the introduction mentioned too many unrelated statements, which should be summarized and expressed in one or two sentences.
2. Line 97-98: The spatial range of this study area is larger than the Taiwan Strait and covers eastern Taiwan which does not belong to the Taiwan Strait. This serious problem should be revised.
3. Line 270-278: This should be moved to 4.2.
4. In 4.1, the authors discussed the spatial distribution of P. niger in the Taiwan Strait with the effects of three major oceanic currents in the Taiwan Strait: The Kuroshio Branch Current, the China Coastal Current, and the South China Sea warm current. However, Jan et al. (2002) pointed out that the activities of the Northeastern monsoon and topography were the important factors controlling the northward intrusion of the Kuroshio Branch and South China Sea warm currents. Furthermore, Kuo et al. (2018) emphasized the considerable effects of topography on the circulation in the Taiwan Strait, especially Chang-Yun Rise. Also, Tang et al. (2002) identified five upwelling zones in the Taiwan Strait and summarized temporal observations of those upwelling with their impacts on the season of fishing grounds. From the reader’s point of view, the authors should explain the seasonality of three currents and the possible impacts on the spatial distribution through time (see Fig. 6 in the manuscript). The reviewer’s suggestion should be added to the manuscript.
5. In terms of the importance of this study, the reader agrees with the authors’ ideas partly, the connection between the role of ensemble habitat modeling with SDGs. As shown in Figure 7, the ensemble modeling could contribute to SDG 14, having a relationship with SDG 1, SDG 2, SDG 12, and SDG 14. Generally, the interactions between SDGs are believed to be possible, however, this manuscript only focused on one species – Black Pomfret Parastromateus niger. Based on the fisheries statistic yearbook in 2022 (downloaded via the website of Taiwan Fishery Agency https://www.fa.gov.tw/view.php?theme=FS_AR&subtheme=&id=21), there were decreasing trends in fisheries production and fisheries values from 2012 to 2021 with the value of 0.2 million tons and 20 billion NTD, respectively. Furthermore, Black Pomfret excludes from the list of 15 significant species in Taiwan (fisheries statistic yearbook, 2022). Therefore, it should be careful and give more powerful argument when explaining the connection between Black Pomfret habitat modeling and other SDGs (SDG 1, SDG 2, SDG 9 and SDG 12).
6. Regarding the discussion on Habitat Modelling Approach for Sustainable Development, the authors believed that habitat modeling could achieve the Black Pomfret habitat identification, contributing to promoting stock sustainability and avoiding overfishing. However, from a personal point of view, the authors should provide more evidence for this statement and explain more details on related fisheries management policy.
7. Because the manuscript focuses on the Black Pomfret habitat modeling in the Taiwan Strait and is not purely theoretical research, part 4.2 and 4.3 do not focus on this species. In the reviewer’s opinion, the authors could emerge two parts and give more related details.
I look forward to receiving your revised manuscript.

References:
1. Jan S, Wang J, Chern CS, Shao SY (2002) Seasonal variation of the circulation in the Taiwan Strait. Journal of Marine systems, 35: 249-268.
2. Kuo Y., Chan J, Wang YC, Shen YL, Chang Y, Lee MA (2018) Long-term observation on sea-surface temperature variability in the Taiwan Strait during the northeast monsoon season. International journal of Remote Sensing, 39(13): pp 4330-4342.

Experimental design

The article is within the aims and scope of the journal with the well-defined the research question, but the minor issues of this manuscript should be revised as the following:
1. To separate references, we use “;” instead of “,” as line 35.
2. The unit of SSC should be mg/m3 (line 20)
3. What are N.CPUE and S.CPUE (Line 131 and 152)? The first time they appear should not be abbreviated.

Validity of the findings

No comment

Additional comments

The article is within the aims and scope of the journal with a well-defined the research question, but the minor issues of this manuscript should be revised as the following:
1. To separate references, we use “;” instead of “,” as line 35.
2. The unit of SSC should be mg/m3 (line 20)
3. What are N.CPUE and S.CPUE (Line 131 and 152)? The first time they appear should not be abbreviated.

---

## Round 0.3 · accepted · Accept

I think that the authors did a good job in addressing the comments raised during the revision and that the manuscript is now ready for publication.